# Transcriptional inhibition after irradiation occurs preferentially at highly expressed genes in a manner dependent on cell cycle progression

**Zulong Chen**[1†], **Xin Wang**[2,3†], **Xinlei Gao**[2,3], **Nina Arslanovic**[1], **Kaifu Chen**[2,3*], **Jessica K Tyler**[1*]

[1]Weill Cornell Medicine, Department of Pathology and Laboratory Medicine, New York, United States; [2]Basic and Translational Research Division, Department of Cardiology, Boston Children's Hospital, Boston, United States; [3]Department of Pediatrics, Harvard Medical School, Boston, United States

**\*For correspondence:**
Kaifu.Chen@childrens.harvard.edu (KC);
jet2021@med.cornell.edu (JKT)

[†]These authors contributed equally to this work

**Competing interest:** The authors declare that no competing interests exist.

**Abstract** In response to DNA double-strand damage, ongoing transcription is inhibited to facilitate accurate DNA repair while transcriptional recovery occurs after DNA repair is complete. However, the mechanisms at play and the identity of the transcripts being regulated in this manner are unclear. In contrast to the situation following UV damage, we found that transcriptional recovery after ionizing radiation (IR) occurs in a manner independent of the HIRA histone chaperone. Sequencing of the nascent transcripts identified a programmed transcriptional response, where certain transcripts and pathways are rapidly downregulated after IR, while other transcripts and pathways are upregulated. Specifically, most of the loss of nascent transcripts occurring after IR is due to inhibition of transcriptional initiation of the highly transcribed histone genes and the rDNA. To identify factors responsible for transcriptional inhibition after IR in an unbiased manner, we performed a whole genome gRNA library CRISPR/Cas9 screen. Many of the top hits on our screen were factors required for protein neddylation. However, at short times after inhibition of neddylation, transcriptional inhibition still occurred after IR, even though neddylation was effectively inhibited. Persistent inhibition of neddylation blocked transcriptional inhibition after IR, and it also leads to cell cycle arrest. Indeed, we uncovered that many inhibitors and conditions that lead to cell cycle arrest in G$_1$ or G$_2$ phase also prevent transcriptional inhibition after IR. As such, it appears that transcriptional inhibition after IR occurs preferentially at highly expressed genes in cycling cells.

## eLife assessment

This **important** work describes a **compelling** analysis of DNA damage-induced changes in nascent RNA transcripts, and a genome-wide screening effort to identify the responsible proteins. A significant discovery is the inability of arrested cells to undergo DNA damage-induced gene silencing, which, is attributed to an inability to mediate ATM-induced transcriptional repression. This work will be of general interest to the DNA damage, repair, and transcription fields, with a potential impact on the cancer field.

## Introduction

DNA double-strand breaks (DSB) are one of the most deleterious types of DNA lesions. Failure to repair a single DSB can lead to loss of a chromosome arm or cell death and inaccurate repair can lead

to changes such as insertions, deletions, and translocations. Accordingly, the cell has developed an intricate DNA damage response (*Jackson and Bartek, 2009*). In vertebrate cells, the DNA damage response is mediated through activation of three PI3-like kinases: ataxia telangiectasia mutated (ATM), Ataxia telangiectasia and Rad3-related (ATR), and DNA-dependent protein kinase (DNA-PK) (*Blackford and Jackson, 2017*), which coordinates DNA repair and the DNA damage cell cycle checkpoint which arrests cells until DSBs are repaired.

Additionally, ATM and DNA-PK have been shown to inhibit transcription in response to DSBs in a manner that occurs so transiently that it can only be detected when examining the nascent, not bulk, transcripts (*Pankotai and Soutoglou, 2013*). The transcription resumes or 'recovers' immediately after DSB repair (*Pankotai and Soutoglou, 2013*). This transient inhibition of transcription after DSBs was initially shown for RNA polymerase I (Pol I) transcripts, where ATM triggered the reduction of nascent ribosomal gene transcripts, shown by visualization of a labeled ribonucleotide analog within the nucleolus, after exposure to ionizing radiation (IR) (*Kruhlak et al., 2007*). Mechanistically, the DSBs triggered a reduction in Pol I initiation complex assembly and led to premature displacement of elongating Pol I from the rDNA genes (*Kruhlak et al., 2007*). Using a reporter that allowed for visualization of repair protein recruitment and local transcription within cells, it was subsequently shown that ATM also mediates the inhibition of RNA polymerase II (Pol II) transcriptional elongation of genes in the vicinity of I-SceI endonuclease-induced DSBs (*Shanbhag et al., 2010*). This transcriptional inhibition was partly dependent on the E3 ubiquitin ligases RNF8 and RNF168, whereas transcriptional recovery depended on the USP16 enzyme that deubiquitylates histone H2A (*Shanbhag et al., 2010*). Additional mechanistic analyses using this same system revealed that ATM-dependent phosphorylation of the ATP-dependent nucleosome remodeler PBAF is required for local transcriptional inhibition of Pol II transcription flanking a DSB (*Kakarougkas et al., 2014*), indicating that chromatin changes are also required for transcriptional inhibition in response to DSBs. The purpose of local transcriptional inhibition is to allow efficient and accurate DSB repair (*Kakarougkas et al., 2014*; *Meisenberg et al., 2019*). Polycomb group proteins and cohesin have also been shown to be required for local transcriptional inhibition of Pol II transcription flanking a DSB, although their role is unclear (*Meisenberg et al., 2019*; *Kakarougkas et al., 2014*), further indicating that chromatin structure and potentially chromosome architecture also regulate transcriptional inhibition in response to DSBs.

Somewhat surprisingly, a distinct mechanism has been reported for the inhibition of Pol II transcription at genes containing a DSB induced by the endonuclease I-PpoI (*Pankotai et al., 2012*). In this case, both Pol II initiation and elongation were reduced adjacent to the DSB, in a manner dependent on DNA-PK and the proteasome (*Pankotai et al., 2012*). Mechanistically, DNA-PK appeared to help recruit the E3 ubiquitin ligase WWP2 to DSBs, which then promoted the proteosome-dependent eviction of Pol II (*Caron et al., 2019*). In the absence of WWP2, the DNA repair machinery was not efficiently recruited, indicating again that the reason for transcriptional inhibition in cis flanking a DSB is to promote DNA repair (*Caron et al., 2019*). The papers examining transcriptional inhibition around DSBs induced by endonucleases generally find the transient repression occurs locally or in cis to the DSB (*Iannelli et al., 2017*). However, another study found induction of the same set of ~200 transcripts soon after irradiation and endonuclease break induction that occurred in a manner dependent on ATM and p53, while only 33 nascent transcripts were down regulated after DSB induction (*Venkata Narayanan et al., 2017*). Yet another study found that more genes were repressed than induced after inducing global DSBs with neocarzinostatin, and this occurred via p53-mediated down-regulation of MYC (*Porter et al., 2017*). As such, there are contradictory findings in the field at present. Furthermore, the mechanism of transcriptional recovery after DSB repair is far from clear. In response to UV damage, global transcriptional inhibition and recovery occurs, and this transcriptional recovery after UV repair is dependent on the histone variant H3.3 histone chaperone HIRA (*Bouvier et al., 2021*; *Adam et al., 2013*). Mechanistically, HIRA functioned to repress the transcriptional repressor ATF3, in turn promoting transcriptional recovery after UV repair (*Bouvier et al., 2021*).

In contrast to the studies to date that have only examined local transcription inhibition occurring in cis after DSB damage, we sought to examine transient transcriptional inhibition after induction of global DSBs by IR exposure, and the subsequent transcriptional recovery after DSB repair, via fluorescent labeling of a ribonucleotide analog incorporated only into nascent transcripts. Unlike the situation following UV repair, we do not find a role for HIRA in transcriptional recovery after DSB repair. Our sequencing of the nascent transcripts after irradiation identified a programmed transcriptional

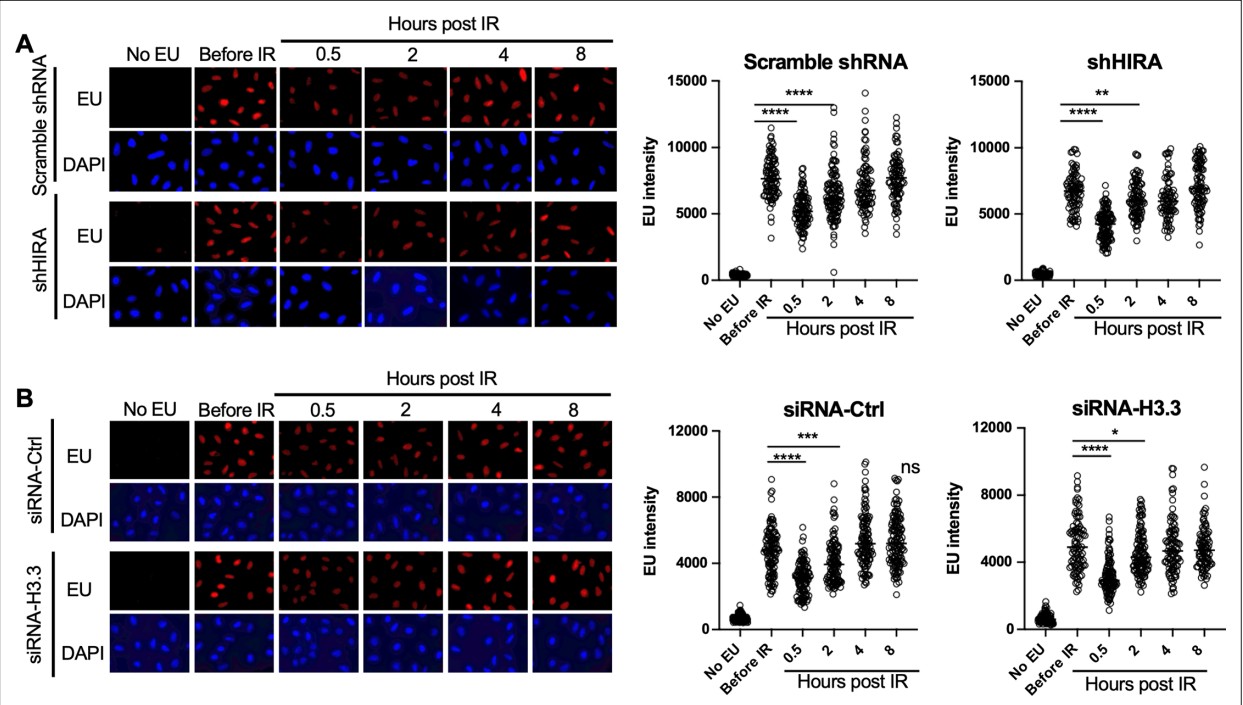

**Figure 1.** Transcriptional inhibition after irradiation and transcriptional restart after DNA repair in a HIRA-independent manner. (**A**) U2OS cells were transfected with either a scrambled shRNA or shRNA against HIRA, and were either incubated with ethynyl uridine (EU) or not, as indicated, and were irradiated (10 Gy) or not as indicated, followed by detection of EU by click chemistry of a fluorophore and DNA was detected by DAPI staining. The right panel shows quantitation of the mean intensity of EU in at least 80 cells for each condition. ****p<0.001, **p<0.01, by student's t-test. (**B**) U2OS cells were transfected with either a control siRNA (siRNA-Ctrl) or two siRNAs against each gene encoding H3.3 (siRNA-H3). EU and DAPI were detected as described in A and quantitated as described in A. ****p<0.001, ***p<0.005, *p<0.05 by student's t-test.

The online version of this article includes the following source data and figure supplement(s) for figure 1:

**Figure supplement 1.** Transcriptional inhibition after irradiation and transcriptional restart after DNA repair in U2OS cells.

**Figure supplement 2.** Confirmation of knockdown of HIRA (**A**) and H3.3 (**B**).

**Figure supplement 2—source data 1.** Original file for the Western blot analysis in *Figure 1—figure supplement 1A* (anti-HIRA and anti-GAPDH).

**Figure supplement 2—source data 2.** PDF containing *Figure 1—figure supplement 1A* and original scans of the relevant Western blot analysis (anti-CUL4A, anti-CUL4B, and anti-GAPDH) with highlighted bands and sample labels.

**Figure supplement 2—source data 3.** Original file for the Western blot analysis in *Figure 1—figure supplement 1B* (anti-H3.3 and anti-GAPDH).

**Figure supplement 2—source data 4.** PDF containing *Figure 1—figure supplement 1B* and original scans of the relevant Western blot analysis (anti-H3.3 and anti-GAPDH) with highlighted bands and sample labels.

program where a larger number of protein-coding genes were upregulated than downregulated. The genes that were immediately downregulated after IR tended to be highly transcribed genes including the rDNAs and histones, while the upregulated genes tended to be transcribed at a lower level. We developed a flow cytometry-based assay of nascent transcripts and used it as the basis for a whole genome gRNA screen to identify factors required for transcriptional inhibition after IR. In addition to finding ATM as being required for inhibition of transcription after DSB induction, we found that depletion of factors leading to cell cycle arrest also blocked transcriptional inhibition.

## Results

### HIRA-independent transcriptional inhibition and recovery after ionizing radiation

To detect bulk changes in nascent transcripts after irradiation in situ, we added the uridine analog ethynyl uridine (EU) to human U2OS cells for 30 min. The incorporated EU was detected by click chemistry to a fluorescent azide followed by immunofluorescence microscopy (*Jao and Salic, 2008*).

A reduction in bulk nascent transcripts was apparent 30 min after exposure to 10 Gray ionizing radiation (IR), and the transcriptional recovery was already occurring 2 hr after irradiation (*Figure 1A*). Co-immunofluorescence with γH2AX showed that the DNA damage signal was greatly reduced at the same time point after IR where transcriptional recovery occurred (*Figure 1—figure supplement 1*), consistent with transcriptional recovery occurring after DSB repair. Given that the histone variant H3.3 histone chaperone HIRA promotes transcriptional recovery after UV repair (*Bouvier et al., 2021*; *Adam et al., 2013*), we tested whether that was also the case for transcriptional recovery after IR. We found that shRNA depletion of HIRA (*Figure 1—figure supplement 2*) had no effect on transcriptional inhibition nor recovery after IR (*Figure 1A*). In agreement, depletion of transcripts encoded from both H3.3 genes (*Figure 1—figure supplement 2*) had no effect on transcriptional inhibition or transcriptional recovery after IR (*Figure 1B*). As such, the requirement for HIRA for transcriptional recovery differs following IR and UV exposure, suggesting differences in the mechanism of these processes.

## Establishment of a flow cytometry-based assay for transcriptional inhibition and recovery after irradiation

Given that the read-out of nascent EU labeled transcripts after irradiation is fluorescence, we established a flow cytometry-based assay to allow us to screen for factors regulating transcriptional inhibition and recovery after IR. We established this assay in a murine Abelson virus-transformed pre-B cell line (termed Abl pre-B cells) (*Bredemeyer et al., 2006*) which are non-adherent and are stably transformed with doxycycline-inducible Cas9 (*Chen et al., 2021*). By flow cytometry analysis, effective incorporation of EU into nascent transcripts was apparent in a manner dependent on ongoing transcription because it was inhibited by the global RNA polymerase inhibitor Actinomycin D (*Figure 2A*). The EU signal had 2 peaks (*Figure 2A*) and we asked whether this reflects cell cycle differences in the cells with higher and lower EU incorporation into the nascent transcripts. We conducted cell cycle analysis by labeling newly synthesized DNA with BrdU and staining DNA contents with FxCycle Violet at the same time as using EU to label nascent transcripts. It was apparent that the peak with less EU was from the $G_1$ phase cells, while the peak with more EU was derived from S and $G_2$ phase cells (*Figure 2B*). To determine whether there was a detectable reduction in EU incorporation by flow cytometry after irradiation, we irradiated cells, waited different lengths of times before EU labeling of nascent RNA (*Figure 2C*). The transcriptional inhibition after irradiation was clearly detectable by flow cytometry as early as 15 min after IR, while transcriptional recovery was complete by 4 hr after IR (*Figure 2D*). Also, the extent of transcriptional repression was similar regardless of whether the IR dose was 2, 5, or 10 Gray (*Figure 2D*, *Figure 2—figure supplement 1*). This is consistent with the possibility that the reduction of nascent transcripts after IR is a programmed/signaling response rather than due to proximity of the genes to the DSB, which would have led to a dose-response.

## Bulk reduction of nascent transcripts after IR is mainly due to a decrease in rDNA and histone gene transcription

To gain a better understanding of transcriptional inhibition after IR, we sought to identify the genes whose transcription was being inhibited after IR. We isolated EU labeled nascent total RNA transcripts 30 min after IR and prior to IR from two independent experiments (*Figure 3—figure supplement 1*) and sequenced the EU-RNA (*Figure 3A*). Prior to isolation of the ER labeled nascent RNA, we added equal amounts of the commercial ERCC spike-in to RNA from the same number of cells. This enabled the subsequent normalization of the total read number from the human genome to total reads from the ERCC control, to detect global changes between samples (*Chen et al., 2015*). We observed that the total read count of nascent transcripts declined after IR (*Figure 3B*). Most of the read counts were due to rDNA transcripts, and the decline in bulk transcripts after IR was mostly due to a significant decline in rDNA transcripts (*Figure 3B*, *Figure 3—figure supplement 2A*). By contrast, the total read count from the protein-coding transcripts significantly increased after IR (*Figure 3B*). Analysis of the protein-coding transcripts showed that the transcripts of 3026 and 1388 protein-coding genes increased and decreased after IR, respectively (*Figure 3C and D*, Supp. File 1). To validate our EU-RNA sequencing results, we performed quantitative RT-PCR to measure the nascent transcript levels of genes that were up- and down-regulated after IR. Consistently, we found rDNA transcripts of *28* S and *18* S were significantly downregulated after IR; while the p53-regulated gene *Cyclin-dependent kinase inhibitor 1* (*Cdkn1a*/p21) was highly induced (*Figure 3—figure supplement 3*). The

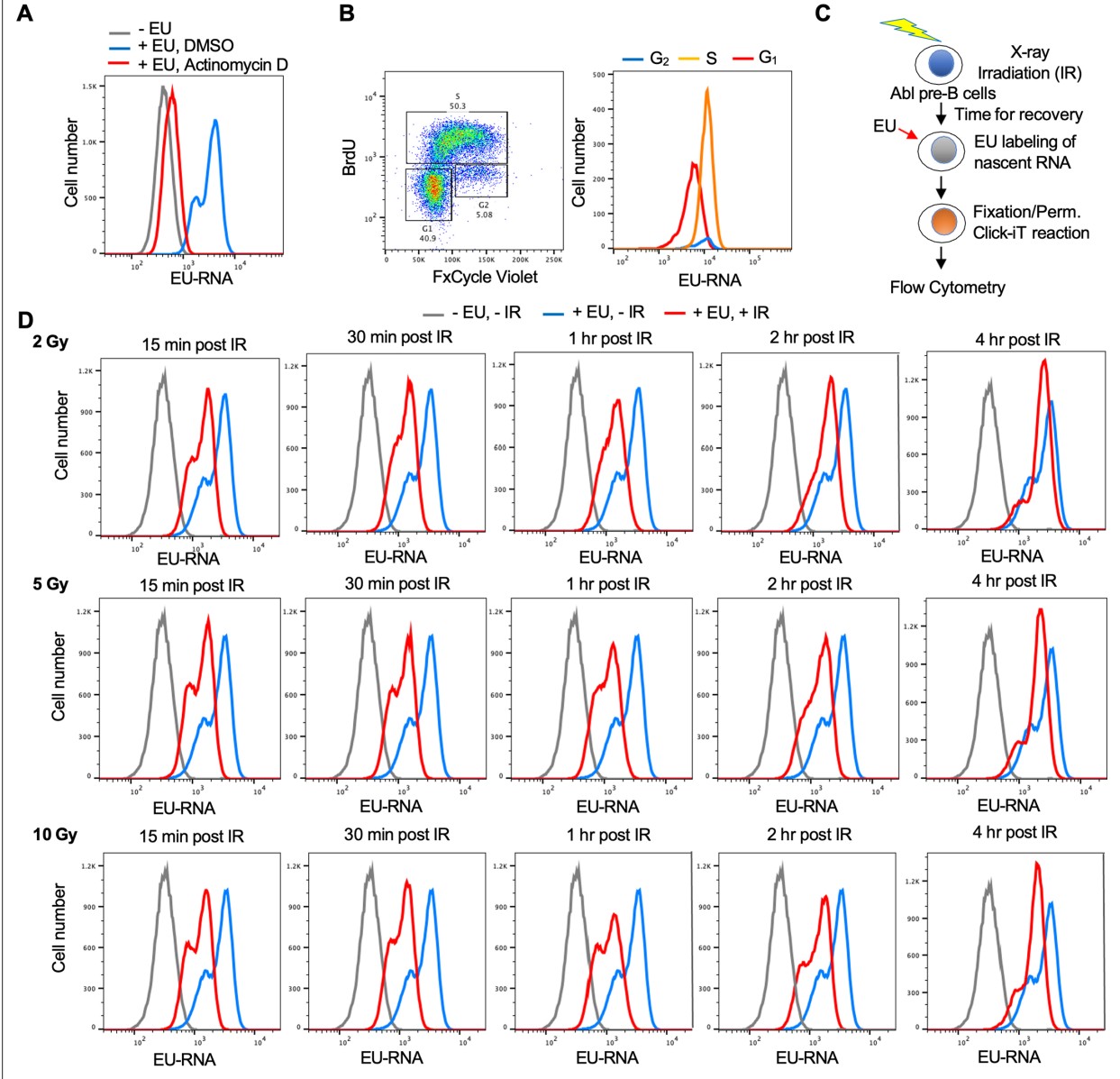

**Figure 2.** Development of a flow cytometry assay for nascent transcripts shows that transcriptional inhibition after ionizing radiation (IR) is not dose-dependent. (**A**) The ethynyl uridine (EU) positive signal in murine Abl pre-B cells detected by flow cytometry is due to transcripts, as indicated by the addition of 5 μM of the general RNA polymerase inhibitor Actinomycin D for 1 hr. (**B**) The two EU peaks observed by flow cytometry correspond to $G_1$ (low peak) and $G_2$ (high peak) cells. Cycling Abl pre-B cells (left panel) were gated for those with 2 N DNA ($G_1$) content or 4 N ($G_2$) content as detected by FxCycle Violate or with BrdU incorporation (S) as indicated and were individually analyzed for EU incorporation into nascent transcripts (right panel). (**C**) Schematic of the assay to detect transcriptional inhibition and transcriptional recovery after IR. (**D**) Time course of transcriptional inhibition and recovery in Abl pre-B cells after IR with the indicated times after IR at the indicated doses of IR.

The online version of this article includes the following figure supplement(s) for figure 2:

**Figure supplement 1.** The mean intensities of the ethynyl uridine (EU) peaks shown in *Figure 2D* are indicated.

gene ontology terms describing the genes that were activated after IR included known DNA damage response pathways and related genes (*Figure 3E*, Supp. File 2). For example, the intrinsic apoptotic signaling pathway in response to DNA damage (GO:0008630), type 2 response (GO:0042092), and cytokine-mediated signaling pathway (GO:0019221) were up regulated significantly. Pro-inflammatory cytokines are the major components of immediate early gene programs, being rapidly activated after irradiation in various cell types (*Schaue et al., 2012*) ultimately leading to radiation-induced fibrosis in cancer patients following radiation therapy (*Kim et al., 2014*; *Yu et al., 2023*). Meanwhile, many

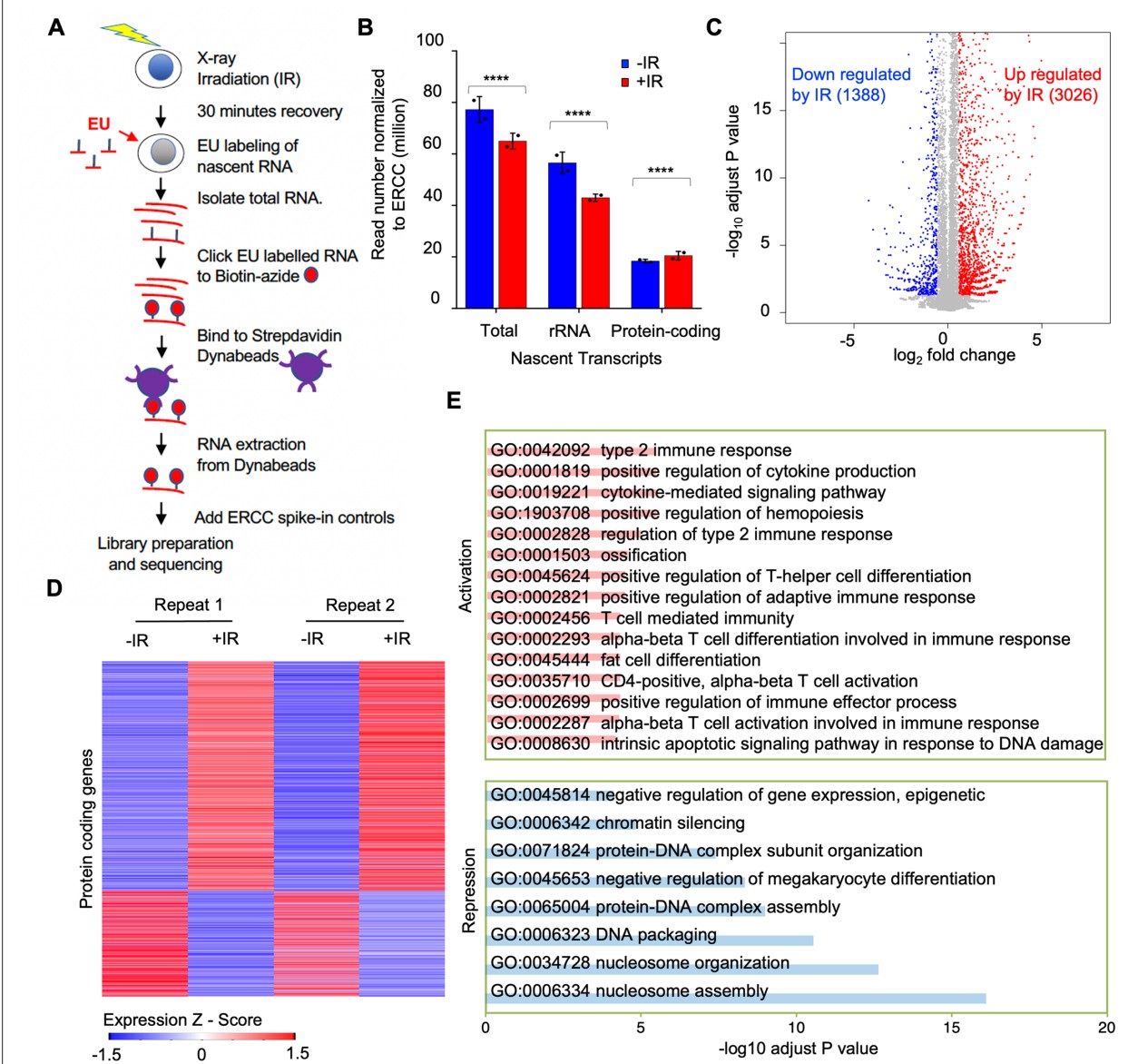

**Figure 3.** Reduction of nascent transcript levels after irradiation is mainly of the rDNA. (**A**) Schematic of nascent transcript sequencing. (**B**) Read counts for the total nascent transcripts, rDNA transcripts, and protein-coding transcripts before and 30 min after ionizing radiation (IR), normalized to ERCC spike-in controls. (**C**) Significantly changed nascent transcripts from protein-coding genes are indicated upon irradiation, and the numbers indicate the number of upregulated and downregulated genes 30 min after IR. Data shown are an average of the two independent experimental repeats. (**D**) Heat map of significantly increased and decreased nascent transcripts 30 min after IR, shown for two independent experimental repeats. Expression z-score was calculated by subtracting the overall average gene abundance from the raw expression for each gene and dividing that result by the standard deviation (SD) of all of the measured counts across all samples. (**E**) Gene ontology analysis of the top significantly enriched GO terms most upregulated after IR (pink) and most downregulated after IR (blue).

The online version of this article includes the following figure supplement(s) for figure 3:

**Figure supplement 1.** Nascent transcript levels before and after 30 min of ionizing radiation (IR).

**Figure supplement 2.** Screen shot from the UCSC browser of nascent transcripts.

**Figure supplement 3.** Validation of nascent transcript levels of differentially expressed genes (DEGs) from ethynyl uridine (EU)-RNA seq by real-time quantitative RT-PCR.

of the genes that were downregulated after IR included gene products that are involved in chromatin organization and nucleosome assembly (*Figure 3E*, *Supplementary file 3*). For example, chromatin silencing (GO:0006342), DNA packaging (GO:0006323), and nucleosome assembly (GO:0006334) genes were down regulated significantly.

To gain more insights on how the transcription of protein-coding genes was regulated after IR, we defined differentially expressed genes (DEGs) between samples before and after IR. We found read number from these DEGs became significantly greater after IR (*Figure 4A*). We sorted the protein-coding DEGs by average expression level of each gene in all four samples: two replicate samples before IR and two replicates after IR. The number of reads derived from the mostly highly expressed protein-coding DEGs became significantly smaller after IR (*Figure 4A and B*). If the gene repression after IR is due to their being in cis to the DNA lesion, it would be expected that genes that were repressed after IR would tend to be longer, because they would be more likely to be damaged. However, this was not the case because the length of the nascent transcripts was equivalent regardless of whether their transcription was repressed, activated, or not changed after IR (*Figure 4C*). Intriguingly, we found that the repressed genes of the top 100 high-expression DEGs tended to be shorter (*Figure 4C*). Next, we inspected the expression level of individual protein-coding genes and confirmed that most changes in gene expression after IR tended to occur for the genes that were activated after IR, while many of the genes that had a high-expression level were repressed after IR, for example, the histone encoding genes (*Figure 4D*). Strikingly, we found that the vast majority of the histone genes showed reduced transcription after IR (*Figure 4E*, *Figure 3—figure supplement 2B*), and was validated by RT-PCR analysis (*Figure 3—figure supplement 3*). Finally, to determine whether transcriptional repression was occurring at the initiation, elongation, or both stages of transcription, we examined the read counts throughout the open reading frames of the repressed protein-coding genes, before and after IR. We found that the decrease of transcripts mainly occurred in the gene body of these genes with similar intensity at both the 3' and 5' ends of the gene body, which indicates transcriptional repression after IR occurred at the initiation stage of transcription (*Figure 4F*). Therefore, these data indicate that the bulk reduction in nascent transcripts after IR is mainly due to reduced transcriptional initiation of the rDNA and histone genes.

## A CRISPR-Cas9 screen identifies ATM, neddylation, and CUL4B as promoting transcriptional inhibition after IR

We performed a genome-wide gRNA library CRISPR-Cas9 screen in Abl pre-B cells, allowing 7 days for the gRNAs to inactivate their target genes (*Figure 5A*). We then sorted the 10% of the cells with the most nascent RNA (high EU) 30 min after IR, as these would include cells with gRNAs corresponding to gene products that are required for transcriptional inhibition after IR. We sequenced the gRNAs within the high EU cells and within the total cell population. We calculated an enrichment score for each of the 5 gRNAs that were included in the library against each gene (Supp. File 4, *Figure 5—figure supplement 1*). *Figure 5B* shows the enrichment scores for the 5 gRNAs for some of the top hits (high EU) from the screen. We found that gRNAs against ATM were enriched in the high EU cells, consistent with the fact that transcriptional inhibition of a gene after induction of a DSB requires ATM signaling (*Shanbhag et al., 2010*). Other top hits included most of the machinery that mediates neddylation, the post-translational covalent addition of NEDD8, a small ubiquitin-like peptide, onto other proteins (*Rabut and Peter, 2008*). These hits included the *Nedd8* gene encoding the NEDD8 ubiquitin-like modifier, *Nae1* encoding NEDD8 activating enzyme E1 subunit 1 NAE1, *Uba3* encoding the Ubiquitin-like modifier activating enzyme 3 UBA3, *Ube2m/Ubc12* encoding the NEDD8-conjugating enzyme UBC12, *Ube2f* encoding a neddylation E2 enzyme UBE2F, and *Rbx1* and *Rbx2* which encode linkers that facilitate NEDD8 transfer from the E2 enzyme to Cullins (*Figure 5B*). We also uncovered gRNAs against the gene encoding the neddylation substrate CUL4B enriched in the high EC cells (*Figure 5B*). These data suggested that neddylation may have a novel role in transcriptional inhibition after IR.

We validated the role of ATM in bulk transcriptional inhibition after IR with the ATM inhibitor Ku55933 (*Figure 5C*). To validate a role for neddylation in transcriptional shut off after IR, we depleted NAE1 with a gRNA to *Nae1* (*Figure 5—figure supplement 2*) and showed that depletion of NAE1 also greatly reduced transcriptional inhibition after IR (*Figure 5D*). Similarly, a 16 hr treatment with the neddylation inhibitor MLN4924 also greatly reduced transcriptional inhibition after IR (*Figure 5E*). To

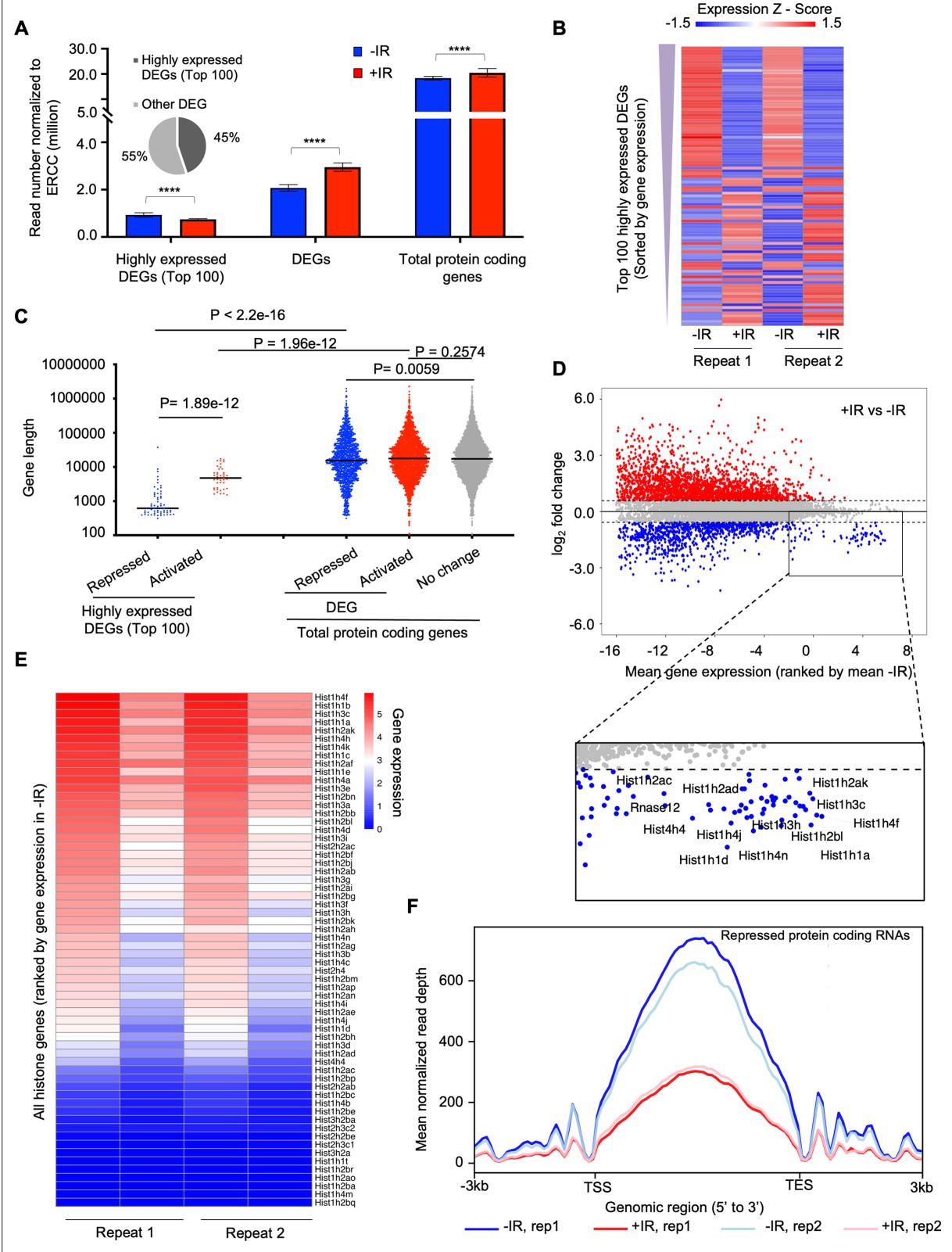

**Figure 4.** The highly transcribed protein-coding genes tend to be repressed after irradiation, due to a decrease in the transcription of the histone genes. (**A**) Plot of the transcript abundance of differentially expressed genes (DEGs) showing that highly expressed genes have reduced nascent transcript levels 30 min after ionizing radiation (IR), while moderately expressed and low-level expressed genes tend to have increased nascent transcript levels 30 min after IR. Mean gene expression and standard deviation are shown in million reads mapped to genes normalized by ERCC spike-in reads.

*Figure 4 continued on next page*

*Figure 4 continued*

Data are shown from two independent experimental repeats (rep) of the experiment. (**B**) Heat map showing nascent transcript levels of the top 100 highly expressed DEGs, ranked by gene expression from top (highest) to bottom, 30 min after IR, shown for two independent experimental repeats. Expression z-score was calculated by subtracting the overall average gene abundance from the raw expression for each gene and dividing that result by the standard deviation (SD) of all the measured counts across all four samples. (**C**) Among the top 100 of highly expressed protein-coding genes, repressed genes are significantly shorter compared to activated genes. The activated, non-changed, and repressed genes show little difference in gene size (the data are averaged for each gene between two independent experimental repeats). (**D**) Plot of change in gene expression after IR against mean gene expression ($log_2$), ranked by mean gene expression in samples before IR on the x-axis, for all nascent transcripts. Some of the highly expressed genes whose nascent transcript levels decreased after IR are labeled in the rectangle, including histone genes. (**E**) Heat map of nascent transcripts of all histone genes before (left) and 30 minutes (right) after IR, shown for two independent experimental repeats. (**F**) The average read counts for repressed protein-coding genes throughout their gene length before and after IR for two independent repeats of the experiment.

ensure that the role of neddylation in transcriptional inhibition after IR is not unique to the murine Abl pre-B cells, we found that transcriptional inhibition, detected by fluorescence microscopy, in U2OS cells was significantly reduced upon inhibition of neddylation (*Figure 5F*). As such, these data suggest that neddylation is required for efficient transcriptional inhibition after IR.

While most neddylation substrates have only been identified upon overexpression of the neddylation machinery, the Cullins have been shown to be *bone fide* neddylation substrates in vivo (*Pan et al., 2004*). Given that we found gRNAs to *Cul4b* encoding CULLIN 4B enriched in the EU high cells (*Figure 5B*) we asked whether its neddylation is induced after IR. We found that when compared to the level of unneddylated CUL4A there was not a significantly higher proportion of neddylated CUL4A or CUL4B in U2OS cells after IR (*Figure 6A*). We used the neddylation inhibitor MLN4924 as a positive control to confirm which bands were neddylated CUL4A and CUL4B (*Figure 6A*). To determine whether CUL4B or CUL4A were required for IR induced transcriptional inhibition, we made stable Abl pre-B cell lines lacking each protein by gRNA-mediated disruption of the *Cul4a* or *Cul4b* genes. Loss of CUL4A had no effect on transcriptional inhibition after IR, while loss of CUL4B partially reduced transcriptional inhibition after IR (*Figure 6B*, *Figure 6—figure supplement 1A*). Given that CUL4B and CUL4A show some functional redundancy (*Hannah and Zhou, 2015*; *Brown et al., 2015*), we depleted both proteins at the same time. Additional transient bulk gRNA transfection-mediated depletion of CUL4A from cells lacking CUL4B (depletion of both CUL4A and CUL4B is lethal) did not further increase the block of transcriptional inhibition after IR (*Figure 6—figure supplement 1B*). As such, these data indicate that CUL4B but not CUL4A contributes to transcriptional inhibition after IR. Given that neddylation of CUL4A/CUL4B regulates cell cycle progression (*Hannah and Zhou, 2015*), and we saw changes in the distribution of the cell cycle phases upon CUL4A/CUL4B depletion and upon blocking neddylation as indicated by the change in relative heights of the EU low ($G_1$) and EU high (S/$G_2$) peaks (*Figures 5D, E and 6B* and *Figure 6—figure supplement 1B and C*), we wondered if cell cycle arrest may be influencing transcriptional inhibition after IR. Accordingly, we inhibited neddylation for 1–3 hr in Abl pre-B cells, which effectively inhibited neddylation (*Figure 6C*), and tested the effect on transcriptional inhibition after IR. Strikingly, we observed transcriptional inhibition after IR even upon neddylation inhibition treatment for 1–3 hr (*Figure 6D*), indicating that neddylation is not required for transcriptional inhibition after IR. We also observed that the length of time required for treatment with the neddylation inhibition to block transcriptional inhibition after IR (*Figure 5E*) caused arrest of the cell cycle in $G_2$ phase (*Figure 6—figure supplement 2*). Therefore, these data indicate that neddylation per se is not required for transcriptional inhibition after IR. They also raise the possibility that the cell cycle arrest caused by persistent loss of neddylation may prevent transcriptional inhibition after IR.

## Cell cycle arrest in $G_1$ or $G_2$ phase blocks inhibition of transcription after IR

To directly determine the relationship between cell cycle arrest caused by prolonged neddylation inhibition and transcriptional inhibition after IR, we labeled the Abl pre-B cells with both EU (nascent transcripts) and 7-Aminoactinomycin D (7-AAD) (DNA stain). Prolonged neddylation inhibition led to accumulation of cells with a 4 N DNA content and greatly reduced transcriptional inhibition after IR (*Figure 7A*, *Figure 6—figure supplement 2*). To investigate if this correlation was specific to neddylation inhibition, we used an unrelated inhibitor that causes cell cycle arrest with a 4 N DNA content,

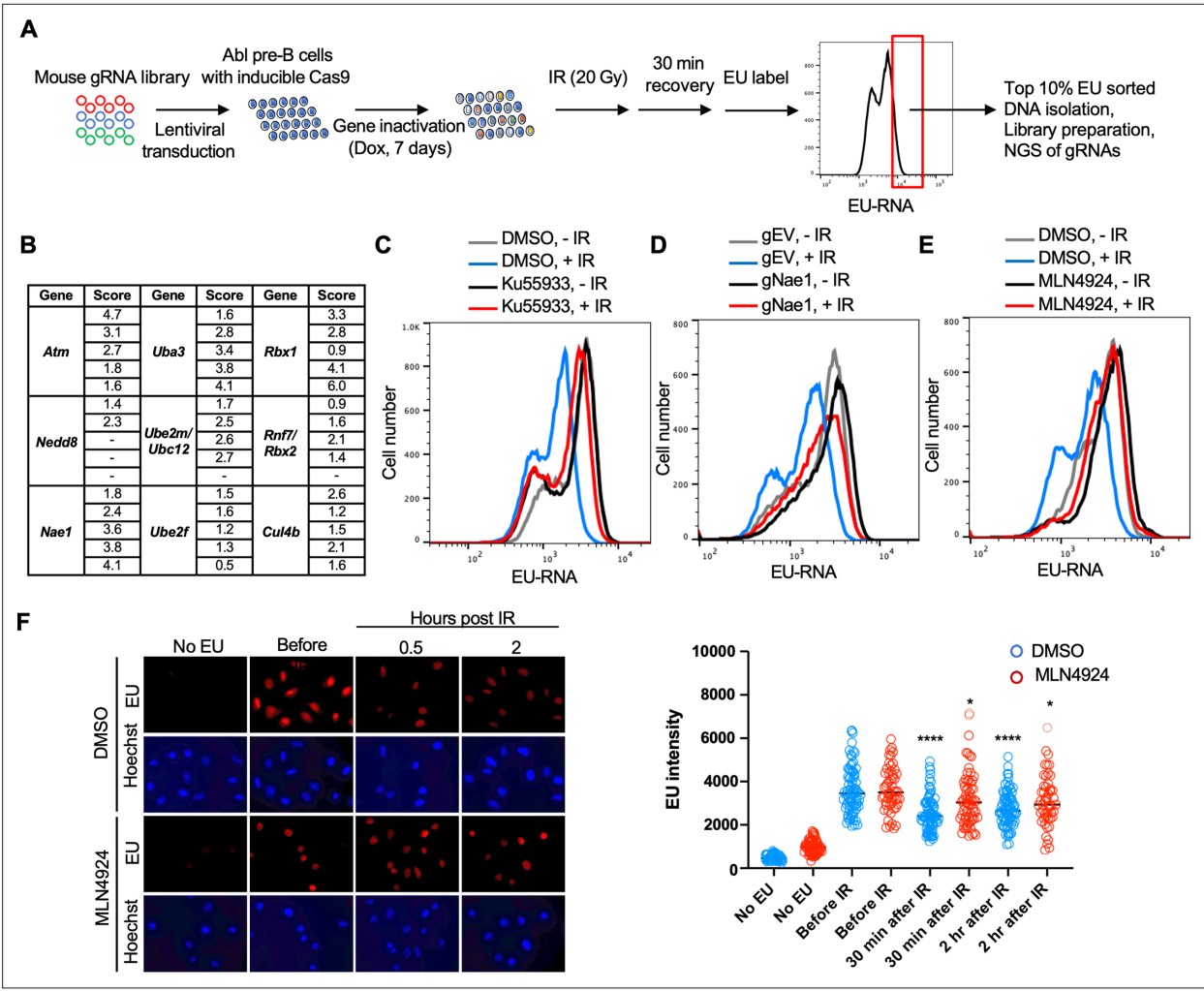

**Figure 5.** Whole genome gRNA screen CRISPR-Cas9 screen for factors involved in transcriptional inhibition after irradiation identifies the neddylation pathway. (**A**) Schematic of whole genome gRNA CRISPR-Cas9 screen for gene products that promote transcriptional inhibition after IR. (**B**) Fold enrichment of five guide RNAs against the indicated genes in the 10% of cells with most ethynyl uridine (EU) incorporated into transcripts 30 min after ionizing radiation (IR). (**C**). Inhibition of ataxia telangiectasia mutated (ATM) greatly reduces transcriptional inhibition 30 min after IR in Abl pre-B cells. The inhibitor was used at 15 µM for 1 hr. (**D**) gRNA-mediated depletion of Nae1 greatly reduces transcriptional inhibition 30 min after IR in Abl pre-B cells. (**E**) Inhibition of neddylation greatly reduces transcriptional inhibition 30 min after IR. The inhibitor was used at 1 µM for 16 hr in Abl pre-B cells. (**F**) Inhibition of neddylation reduces transcriptional inhibition after IR in U2OS cells, as detected by fluorescence analysis of nascent transcripts as described in legend to *Figure 1* and quantitated as in *Figure 1*. Significant differences after IR compared to before IR are indicated by asterisks, where ****$p<0.001$, *$p<0.05$ by student's t-test. All experiments in this figure are in murine Abl pre-B cells.

The online version of this article includes the following source data and figure supplement(s) for figure 5:

**Figure supplement 1.** CRISPR-Cas9 screen identifies genes promoting transcriptional inhibition after ionizing radiation (IR).

**Figure supplement 2.** Confirmation of knockdown of Nae1.

**Figure supplement 2—source data 1.** Original file for the Western blot analysis in *Figure 5D* (anti-Nae1 and anti-GAPDH).

**Figure supplement 2—source data 2.** PDF of Western blot analysis in *Figure 5D* and original scans of the relevant Western blot analysis (anti-Nae1 and anti-GAPDH) with highlighted bands and sample labels.

RO-3306, a CDK1 inhibitor (*Vassilev et al., 2006*). We also saw accumulation of cells with a 4 N DNA content and no transcriptional inhibition after IR upon CDK1 inhibition (*Figure 7B*, *Figure 6—figure supplement 2*). To determine whether this effect was unique to cells arrested with a 4 N DNA content or was shared with conditions that cause arrest in $G_1$ phase, we treated cells with 10% or 0.1% FBS, where the later lead to arrest in $G_1$ phase (due to serum depletion) and prevented transcriptional inhibition after IR (*Figure 7C*). It is relevant to point out that the level of bulk nascent transcripts in

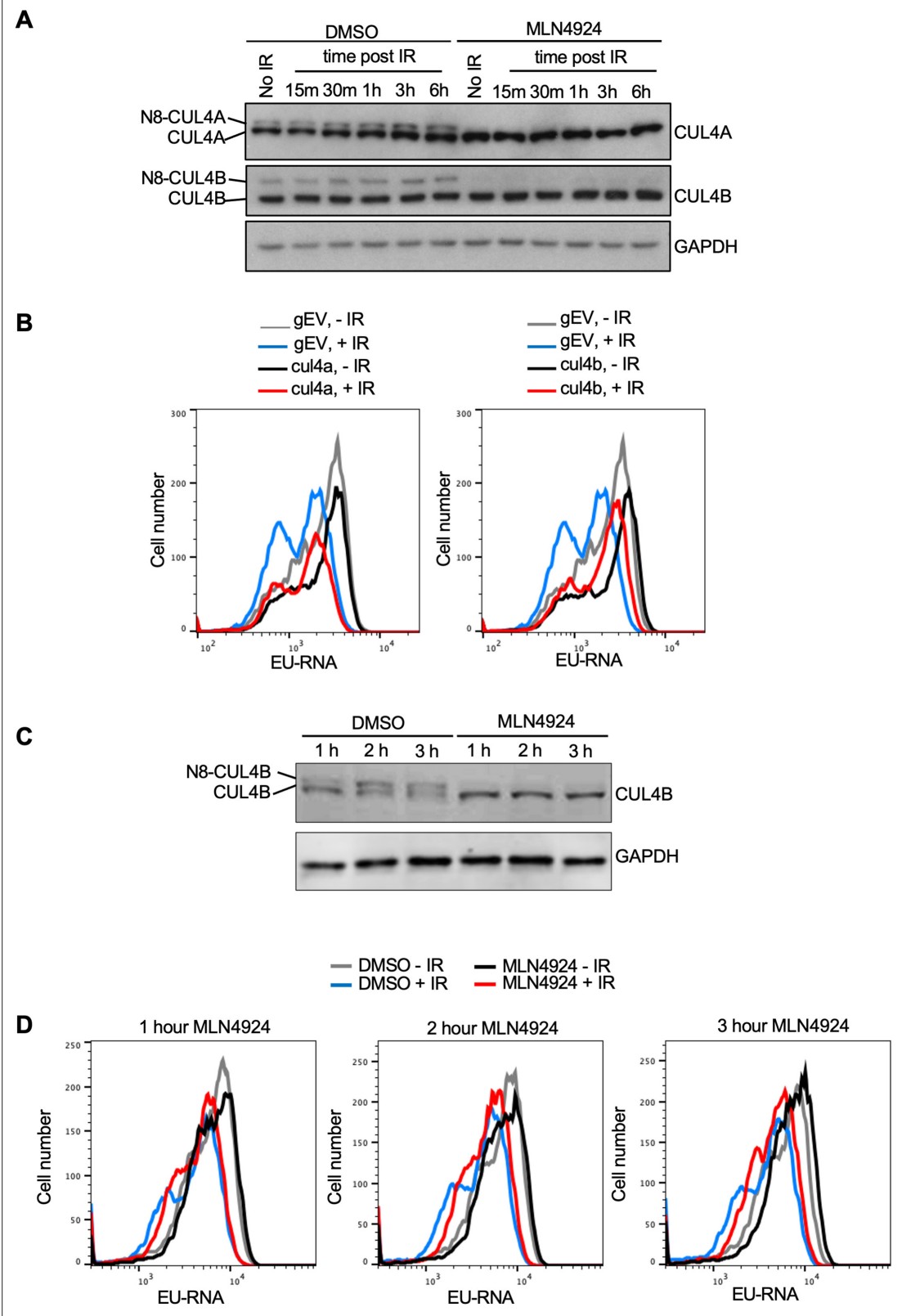

**Figure 6.** CUL4B but not CUL4A contributes to transcriptional inhibition after irradiation. (**A**) Analysis of CUL4A and CUL4B neddylation in U2OS cells after ionizing radiation (IR), in the absence of presence of 10 μM treatment for 3 hr with the neddylation inhibitor. N8 indicates the neddylated species. (**B**) Analysis of nascent transcripts in Abl pre-B cell lines stably depleted of CUL4A or CUL4B 30 min after IR, as indicated. (**C**) Short treatment of Abl

*Figure 6 continued on next page*

*Figure 6 continued*

pre-B cells with neddylation inhibitor MLN4924 is sufficient to block neddylation of CUL4B. (**D**) The same cells used in C were analyzed for ethynyl uridine (EU) incorporation into nascent transcripts 30 min after irradiation, without or with the indicated time of MLN4924 treatment before irradiation.

The online version of this article includes the following source data and figure supplement(s) for figure 6:

**Source data 1.** Original file for the Western blot analysis in *Figure 6A* (anti-CUL4A, anti-CUL4B, and anti-GAPDH).

**Source data 2.** PDF containing *Figure 6A* and original scans of the relevant Western blot analysis (anti-CUL4A, anti-CUL4B, and anti-GAPDH) with highlighted bands and sample labels.

**Source data 3.** Original file for the Western blot analysis in *Figure 6C* (anti-CUL4B and anti-GAPDH).

**Source data 4.** PDF containing *Figure 6C* and original scans of the relevant Western blot analysis (anti-CUL4B and anti-GAPDH) with highlighted bands and sample labels.

**Figure supplement 1.** Analysis of CUL4A and CUL4B depletion.

**Figure supplement 1—source data 1.** Original file for the Western blot analysis is in *Figure 6—figure supplement 1A* (anti-CUL4A, anti-CUL4B, and anti-GAPDH).

**Figure supplement 1—source data 2.** PDF containing *Figure 6—figure supplement 1A* and original scans of the relevant Western blot analysis (anti-CUL4A, anti-CUL4B, and anti-GAPDH) with highlighted bands and sample labels.

**Figure supplement 2.** Cell cycle analysis of cells treated with MLN4924 and RO3306.

cells arrested in $G_1$ by serum depletion is still far higher than cells lacking EU (*Figure 7C*), suggesting that the limited transcriptional inhibition after IR in $G_1$ arrested cells is not just because there is only minimal transcription occurring. In agreement, two CDK4/6 inhibitors, Ribociclib and Palbociclib, that lead to $G_1$ arrest also prevented transcriptional inhibition after IR (*Figure 7D and E*). Intriguingly, these data indicate that multiple different treatments that lead to cell cycle arrest in $G_1$ or $G_2$ per se prevent transcriptional inhibition after IR.

To gain molecular insight into why ATM-dependent bulk transcriptional inhibition occurs after IR in cycling cells but not in arrested cells, we investigated the possibility that ATM-dependent pathways that inhibit rDNA transcription after IR in cycling cells (*Kruhlak et al., 2007*) fail in arrested cells. In cycling cells, DSBs within the rDNA trigger the formation of nucleolar caps due to damaged rDNA and associated proteins relocalizing to the nucleolar periphery (*van Sluis and McStay, 2017*). In cycling cells, DSBs within the rDNA trigger ATM-dependent phosphorylation of Treacle which promotes recruitment of NBS1 and TOPBP1 to the nucleolar caps to inhibit rDNA transcription (*Larsen et al., 2014*; *Mooser et al., 2020*). Accordingly, in cycling cells, TOPBP1 accumulates in the nucleolar caps after induction of DSBs, concomitant with repression of rDNA transcription (*Sokka et al., 2015*; *Mooser et al., 2020*). We asked whether TOPBP1-eGFP recruitment to nucleolar caps was disrupted after DNA damage in cells arrested by treatment with MLN4924 or RO-3306. The overall induction of TOPBP1 expression was not affected by inhibitor treatment (*Figure 7—figure supplement 1*). However, the percentage of cells with TOPBP1-eGFP localizing to nucleolar caps after IR was markedly reduced upon MLN4924 and RO-3306 treatment (*Figure 7F and G*). These results are consistent with arrested cells failing to repress rDNA expression after global DNA damage, as a consequence of compromised ATM-dependent localization of TOPB1 to nucleolar caps.

## Discussion

We find that while the bulk abundance of nascent transcripts is rapidly reduced after IR, more protein-coding genes are induced than inhibited after IR. Instead, the reduction in bulk nascent transcript levels that occurs after IR is due to reduced transcriptional initiation of a subset of genes that are the most highly expressed in the cell – the rDNA and histone encoding genes. Notably, bulk transcriptional inhibition after IR did not occur in cells arrested in $G_1$ or $G_2$ phases of the cell cycle indicating that cells need to be cycling for IR to rapidly inhibit bulk transcription. The length-independent and dose-independent reduction in bulk abundance of nascent transcript after IR (*Figure 4C*, *Figure 2D*) suggests that the reduced bulk abundance of nascent transcripts after IR may occur in trans as a programmed event. This is in contrast to studies that have found transcriptional inhibition in cis of a gene immediately adjacent to an endonuclease-induced DSB. Our work indicates that the genome-wide transcriptional response to DSBs after IR cannot be extrapolated from single gene studies.

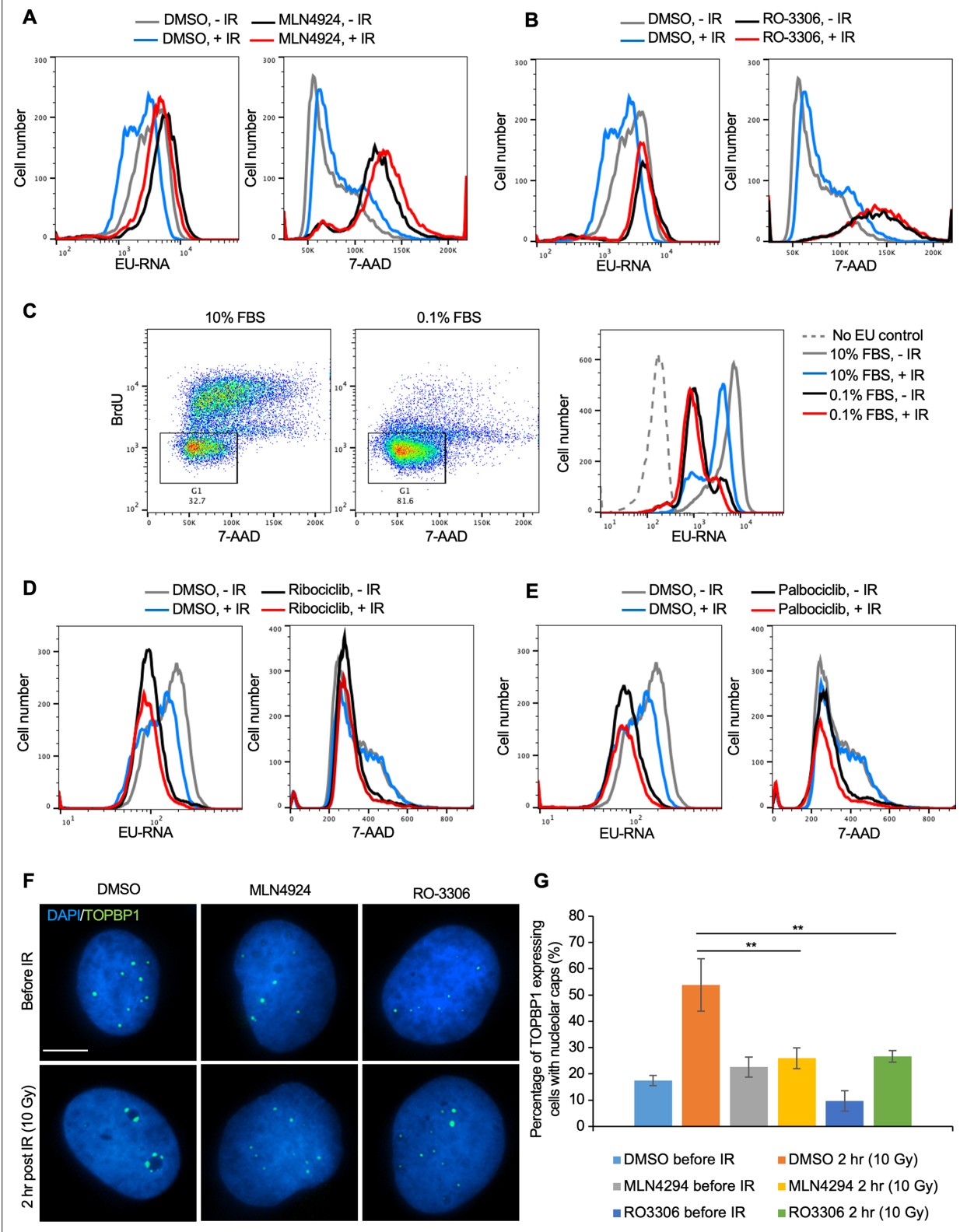

**Figure 7.** Cell cycle arrest in $G_1$ or $G_2$ prevents transcriptional inhibition after DNA damage. (**A**) Abl pre-B cells were treated with MLN4924 (1 µM) for 16 hr, followed by ionizing radiation (IR) and staining of DNA with 7-Aminoactinomycin D (7-AAD) and nascent transcripts with ethynyl uridine (EU) 30 min after IR. (**B**) Abl pre-B cells were treated with RO-3306 (10 µM) for 16 hr, followed by IR and staining of DNA with 7-AAD and nascent transcripts with EU 30 min after IR. (**C**) The left two panels show the cell cycle distribution of Abl pre-B cells after growth for 72 hr in 1% FBS or 0.1% FBS. The

*Figure 7 continued on next page*

*Figure 7 continued*

rectangles and numbers indicate the % of cells with a 2 N DNA content. The right panel shows the EU incorporated into nascent transcripts 30 min after IR for the same samples. (**D**) Abl pre-B cells were treated with Ribociclib (5 µM) for 24 hr followed by IR and staining of DNA with 7-AAD and nascent transcripts with EU 30 min after IR. (**E**) Abl pre-B cells were treated with Palbociclib (5 µM) for 24 hr, followed by IR and staining of DNA with 7-AAD and nascent transcripts with EU 30 min after IR. (**F**) The U2OS cells were treated with doxycycline (1 µg/mL) for 12 hr to express eGFP-TOPBP1. Then, DMSO, MLN4924, and RO3306 were added to the cells for another 16 hr. TOPBP1 localization was shown in cells before IR or 2 hr after IR (10 Gy). Scale bar is 10 µm. (**G**) Quantification of TOPBP1 expressing cells with nucleolar caps before and after IR (10 Gy) in cells treated with MLN4924 and RO-3306. Data shown are an average of the three independent experimental repeats. Significant differences are indicated by asterisks, where \*\*p<0.01 by student's Tt-test.

The online version of this article includes the following figure supplement(s) for figure 7:

**Figure supplement 1.** Quantification of cells with TOPBP1 expression before and after ionizing radiation (IR) (10 Gy) in cells treated with MLN4924 and RO-3306.

Analysis of nascent transcripts at early times after irradiation revealed a different transcriptional response compared to changes in total mRNAs after irradiation (*Lieberman et al., 2017*). These changes in mRNA levels typically occur in an IR dose-dependent manner. By contrast, the bulk changes in nascent transcripts occurred in a dose-independent manner (*Figure 2D*). Analysis of total mRNAs changes following IR showed altered expression of genes involved in signal transduction, regulation of transcription, and metabolism (*Su et al., 2004*). Similarly, the upregulated nascent transcript changes identified pathways including signal transduction, while nascent transcripts from protein-coding genes that affected nucleosome assembly and chromatin structure (histones) were downregulated after IR (*Figure 3B and E*). Finally, most of the gene expression changes detected by analysis of total mRNA continued to increase or decrease over long periods of time, up to 48 hr (*Su et al., 2004*), whereas the reduction of bulk nascent transcript levels occurred in a very transient manner and was already returning to normal by 4 hrs after IR (*Figure 1*, *Figure 2D*).

How is bulk nascent transcription being inhibited after irradiation? Reminiscent of the ATM dependence on the changes in mRNA levels after IR (*Artuso et al., 1995*), the reduction in bulk nascent transcript levels after IR was also dependent on ATM (*Figure 5C*). Given that most of the reduction in bulk nascent transcript levels was due to the rRNA (*Figure 3B*), this is consistent with the previous report of ATM-dependent inhibition of RNA Polymerase I transcription in response to DSBs (*Kruhlak et al., 2007*). In this case, the Pol I transcription appeared to be inhibited at both the initiation and elongation stages (*Kruhlak et al., 2007*). Sequencing analysis of nascent transcripts of protein-coding genes whose expression significantly decreased 30 min after IR suggested that it is the initiation of Pol II transcription that is inhibited, given that the reduction in sequencing reads at the 5' end and 3' ends of open reading frames was equivalent (*Figure 4F*). Studies that examined the mechanism of Pol II transcriptional inhibition of one gene adjacent to an endonuclease induced DSB identified a reduction in transcriptional elongation as indicated by reduced Pol II Ser2 phosphorylation (*Shanbhag et al., 2010*), while other studies found a defect in both Pol II initiation and elongation of a different gene adjacent to an endonuclease induced DSB, in a manner dependent on DNA-PK and the proteasome (*Caron et al., 2019*; *Pankotai et al., 2012*). Our data is consistent with the possibility that the major mechanism for the repression of the ~1000 protein-coding genes after IR is at the transcriptional initiation stage. However, our data do not rule out that transcriptional elongation may be additionally repressed after IR, but would not be observed in our analyses due to the repression of transcriptional initiation.

Rapid inhibition of transcription in cis has been observed following endonuclease-mediated DSB induction, where a DSB-proximal transcriptional reporter was inhibited while a second transcriptional reporter inserted elsewhere in the genome without a proximal FokI nuclease site was not inhibited after induction of Fok1 (*Shanbhag et al., 2010*). If the transcriptional inhibition after IR that we observe was occurring in cis, we would expect that longer genes would be more inhibited than shorter genes after IR, as they are more likely to experience a DSB induced by IR. However, this was not the case because the length of the nascent transcripts was equivalent regardless of whether their levels were increased, didn't change, or were decreased after IR (*Figure 4C*). We would also expect that bulk transcript levels would be more reduced with a higher dose of IR if it occurred in cis, but that was not the case (*Figure 2D*). In fact, transcriptional repression around a nuclease-induced DSB can spread hundreds of kb away from the break, throughout a whole topological associated domain marked by gamma

H2AX (*Purman et al., 2019*). Importantly, our data do not contradict that DSBs can induce transcriptional inhibition in cis, rather it likely reflects the random nature of the DNA damage induced by IR is not sufficient to detect inhibition in cis, as every cell will have DNA DSBs at different locations. We also are examining the effect following global DNA damage-induced IR versus the effect of induction of a single or limited number of DSBs by endonuclease induction. It is noteworthy that the nascent transcript levels of more genes rapidly increased after IR than were reduced (*Figures 3 and 4*), such that the situation, at least after IR, is more complex than transcriptional inhibition in cis to the DSBs.

The change in transcript levels after irradiation tended to depend on the expression level of the genes before irradiation. Those genes that were normally most highly transcribed were repressed after IR, while genes that were normally expressed at intermediate or low levels tended to be induced after IR (*Figure 4A*). The mechanistic reason for this is unclear. Among the genes that were most repressed after IR treatment were many of the histone-encoding genes (*Figure 4D and E*). Histone gene expression has been shown to be reduced after IR previously in a manner dependent on ATM and p53, as seen at the mRNA level 10 hr or more after irradiation (*Su et al., 2004*). By contrast, reduction in nascent histone transcript levels occurred 30 min after IR (*Figure 4D and E*). Interestingly, we observed a tremendous expression induction of *Cdkn1a/p21* gene, which encodes a potent cyclin-dependent kinase inhibitor, after IR (*Figure 3—figure supplement 3*). The histone transcriptional coactivator NPAT induces histone transcription when it is phosphorylated by cyclin E/CDK2 (*Zhao et al., 1998*; *Zhao et al., 2000*; *Ma et al., 2000*). As such, highly elevated levels of Cdkn1a/p21 after IR might inactivate cyclin E/CDK2, leading to hypo-phosphorylation of NPAT and an immediate repression of histone gene expression after IR. It also may be relevant that the loci found to be repressed by bulk IR are highly repetitive gene arrays that tend to form nuclear sub-compartments (nucleoli, histone bodies). As such, their likelihood of repeats being in the vicinity of a repeat with an IR-induced DNA lesion in three-dimensional space is high, which may promote their transcriptional repression after IR in trans. Moreover, silencing may spread through the relevant nuclear sub-compartments, consistent with the formation of DNA damage compartments described recently (*Arnould et al., 2023*).

In response to UV exposure, bulk inhibition of transcription occurs, followed by transcriptional recovery after repair of UV-induced damage, where transcriptional recovery after repair of UV-induced damage is dependent on the histone variant H3.3 histone chaperone HIRA (*Bouvier et al., 2021*; *Adam et al., 2013*). Mechanistically, HIRA functioned to repress the transcriptional repressor ATF3, in turn promoting transcriptional recovery after repair of UV-induced damage (*Bouvier et al., 2021*). We find that neither HIRA nor H3.3 is required for recovery of bulk nascent transcripts after DSB repair (*Figure 1*). Additionally, we find that the genes that have altered levels of nascent transcripts after UV damage (*Bouvier et al., 2021*) and after IR are quite distinct (data not shown). However, the UV studies were performed on nascent mRNA from a human cell line, while our studies were performed on total nascent RNA from a mouse cell line.

Why didn't our CRISRP/Cas9 screen of factors responsible for bulk nascent transcript inhibition after IR identify previously reported factors involved in Pol II transcriptional inhibition proximal to an endonuclease-induced DSB? This is likely because most of the reduction in bulk nascent transcript level that we were detecting after IR was due mainly to reduced transcript abundance from the rDNA, rather than Pol II transcripts (*Figure 3B*). We did find ATM in the screen (*Figure 5B and C*), and this is consistent with the fact that ATM is required for rDNA transcriptional inhibition after IR (*Kruhlak et al., 2007*). Many of the most significant hits from the screen encoded factors that are involved in the neddylation pathway and the neddylation substrate CUL4B (*Figure 5D, E and F* and *Figure 6B*). However, short times of neddylation inhibition were sufficient to inhibit neddylation but did not prevent transcriptional inhibition after IR (*Figure 6C and D*). Longer times of neddylation inhibition did prevent transcriptional inhibition after IR (*Figure 5E and F*), but also caused cell cycle arrest (*Figure 7A*, *Figure 6—figure supplement 2*). These results lead us to speculate that neddylation is not directly involved in transcriptional inhibition after IR, but that neddylation promotes cell cycle progression, and that it is the cell cycle arrest occurring upon neddylation inhibition that prevents the bulk reduction of nascent transcript levels after IR. Consistent with the idea that cell cycle arrest per se may be preventing transcriptional inhibition after IR, we also found that CDK1 inhibition leading to $G_2$ arrest, and serum starvation and CDK4/6 inhibition leading to $G_1$ arrest, also prevented bulk transcriptional inhibition after IR (*Figure 7*).

Why would cell cycle arrest in $G_1$ or $G_2$ phases of the cell cycle prevent transcriptional repression of rDNA and histone genes after IR? Transcription of rDNA is known to be reduced in non-cycling cells when total transcript levels were measured (*Moss and Stefanovsky, 1995*; *O'Mahony et al., 1992*) while transcription of histone genes requires ongoing DNA replication (*Sittman et al., 1983*). As such, one possibility for failure to see bulk reduction in nascent transcript abundance after IR in arrested cells may be that rDNA and histone transcription are already reduced in $G_1$ and $G_2$ arrested cells. However, we found that the total level of bulk nascent transcripts during $G_2$ arrest before IR were equivalent to, or more than, the level of bulk nascent transcripts before cell cycle arrest (compare gray (no arrest) to black lines (cell cycle arrest) in *Figures 5C–E, 7A and B*) which suggest that the levels of rDNA transcripts may not be reduced during $G_2$ arrest in our experiments given that most of the nascent transcripts are from the rDNA in the conditions of our experiments (*Figure 7B*). It is also possible that a factor that is required for transcriptional inhibition of the rDNA after IR is absent or inactive in arrested cells.

In addition to ATM, NBS1 was previously shown to be required for transcriptional inhibition of rDNA after IR (*Kruhlak et al., 2007*). The mechanism for this was unclear, but it is tempting to speculate that the requirement of ATM for transcriptional inhibition after IR in cycling cells may be mediated through ATM-dependent phosphorylation of Treacle. In this form, Treacle functions with TOPBP1 to promote recruitment of NBS1 to nucleolar caps to repress rDNA transcription (*Larsen et al., 2014*; *Mooser et al., 2020*). That TOPB1 fails to relocalize to nucleolar caps after IR in arrested cells (*Figure 7F and G*) is consistent with a potential loss of ATM-dependent treacle phosphorylation in arrested cells, which would prevent reduction in rDNA transcription after DNA damage. Future experiments will reveal further insight into the cell cycle-dependent control of transcriptional inhibition of highly transcribed genes after DNA damage.

# Materials and methods
## Cell culture and transfections
U2OS cells (ATCC, HTB-96) were cultured in McCoy's 5 A (Corning, 10050CV) medium supplemented with 10% fetal bovine serum (FBS) and 1% Penicillin-Streptomycin. Abelson virus-transformed pre-B cells were maintained in DMEM (Thermo Fisher, 11960–077) supplemented with 10% FBS, 1% Penicillin-Streptomycin, 1 x nonessential amino acids, 1 mM sodium pyruvate, 2 mM L-glutamine, and 0.4% beta-mercaptoethanol. HEK-293T cells were maintained in DMEM (Corning, 10–013 CM) supplemented with 10% FBS and 1% Penicillin-Streptomycin. All the cells were grown at 37 °C under a humidified atmosphere with 5% CO2.

SiRNA oligos against human H3F3A and H3F3B (SMARTPool) for RNAi in U2OS cells were purchased from Horizon Discovery (Dharmacon). 100 nM of H3F3A and H3F3B were mixed with Lipofectamine RNAiMAX transfection reagent (Thermo Scientific, 13778150) according to the manufacturer's protocol to knockdown H3.3 for 48 hr. The siRNA control (ON-TARGETplus non-targeting) was also purchased from Horizon Discovery (Dharmacon) and used as a negative control. shRNA lentiviral plasmids against HIRA (5′-TAGAGCATACCAAGATGCC-3′) and the control were described in a previous study (*Huang et al., 2018*). HEK-293T cells were transfected with a mixture of shRNA plasmids and the viral packaging and envelope vectors, pCMV-dR8.2 and pCMV-VSVG. The media containing shRNA virus particles were collected 48–72 hr after transfection and filtered through a 0.45 µm filter. Cells were incubated with the lentiviral supernatant containing 5 µg/ml polybrene (Sigma-Aldrich, S2667) for 24 hr, followed by 1 µg/mL Puromycin selection for another 48 hr. To inactivate Nae1, Cul4a, and Cul4b in bulk cell populations, guide RNAs (gRNAs) against each gene were cloned into pKLV-U6gRNA-EF(BbsI)-PGKpuro2ABFP (Addgene, #62348) modified to express human CD2 or Thy1.1 as cell surface markers. The pKLV-gRNAs lentiviruses were prepared in 293T cells as described above. The Abl pre-B cells containing pCW-Cas9 (addgene, #50661), which can express Cas9 with doxycycline induction, were mixed with viral supernatant supplemented with 5 µg/ml polybrene and centrifuged at 1800 rpm for 1.5 hr at room temperature. After the spin-infection, the transduced cells were maintained in DMEM with 3 µg/ml doxycycline (Sigma-Aldrich, D9891) for 3 days before flow cytometric cell sorting based on hCD2 or Thy1.1 expression. To make stable cell lines depleted of Cul4a and Cul4b, serial dilution of the sorted cells into a 96-well plate was used to

isolate single cells. Western blot analysis was used to determine the knockdown efficiency of each target gene.

In EU flow cytometric and immunofluorescence analysis, final concentrations of 15 µM ATM inhibitor Ku55933 (Selleck Chemicals, S1092) and 5 µM of Actinomycin D (Sigma-Aldrich, A9415) were added to cell culture 1 hr prior to irradiation. 1 µM and 10 µM of neddylation inhibitor MLN4924/Pevonedistat (Active Biochem, A-1139) were used for long (16 hr) and short (1–3 hr) treatments, respectively. To arrest cells in the $G_1$ cell cycle phase, cells were incubated in media supplemented with 5 µM Palbociclib (Selleck Chemicals, S1116) or 5 µM Ribociclib (Selleck Chemicals, S7440) for 24 hr. For serum starvation, Abl pre-B cells were grown in a complete medium containing 10% of FBS to the desired density and collected and washed in a medium with a reduced concentration (0.1%) of FBS. The cells were maintained in the medium with reduced FBS for 72 hr to arrest in the $G_1$ phase. Representative data are shown for experiments repeated three of more times with consistent results.

U2OS cell lines were authenticated by STR profiling, and MCF10A and murine cell lines tested negative for mycoplasma contamination.

## Western blots

The following antibodies were used for western blot: CUL4A (Cell Signaling Technology, 2699 S, 1:1000), CUL4B (Proteintech, 12916–1-AP, 1:1000), H3.3 (Millipore Sigma, 09–838, 1:1000), HIRA (Abcam, ab20655, 1:1000), NAE1 (Thermo Fisher, PA5-59836, 1:500), GAPDH (Sigma-Aldrich, G8795, 1:5000). Representative data are shown for experiments repeated three of more times with consistent results.

## Fluorescence microscopy

For immunofluorescence, the Click-iT RNA Alexa Fluor Imaging Kit (Thermo Fisher, C10330) was used to label newly synthesized RNAs in the cells. Briefly, 50,000 U2OS cells grown on cover slips in a 24-well plate were irradiated with 10 Gray and allowed to recover for indicated times at 37 °C with 5% CO2. 0.5 mM EU was added to the medium and incubated for 45 min for EU incorporation. Cells were then washed with PBS, fixed in 4% paraformaldehyde PBS for 15 min at room temperature, and permeabilized in cold 0.5% Triton X-100 PBS for 10 min. Cells were blocked in 3% BSA-PBS for 1 hr at room temperature and subsequently incubated overnight at 4 °C in primary antibody (anti-γH2AX (S139), EMD Millipore, 05–636). Coverslips were then washed 3 x with PBST (0.05% Tween 20), and incubated with a secondary antibody diluted in 3% BSA PBS (Alexa Fluor 488 Goat anti-mouse IgG, BioLegend, 405319) in the dark for 1 hr at room temperature and washed 3 x with PBST. Click-iT reaction cocktail was prepared according to the manufacturer's protocol and immediately added to the cells to perform click reaction in the dark for 30 min at room temperature. After washes with Click-iT reaction rinse buffer (Component F) and PBS, cells were stained with Hoechst (1:2000) or DAPI (Sigma-Aldrich, D9542) in PBS for 10 min and mounted in Prolong Gold Antifade Mountant (Life Technologies, P-36930). Images were taken on Biotek Lionheart Automatic Microscope and EU intensity quantification was conducted using Biotek Gen5 software. For eGFP-TOPBP1 fluorescence microscopy, 50,000 cells grown on coverslips in 24-well plates were treated with doxycycline (1 µg/mL) for 12 hr to express eGFP-TOPBP1. MLN4924 (1 µM) and RO3306 (10 µM) were added to the cell culture, and the cells were incubated for another 16 hr followed by IR (10 Gy) and recovery for 2 hr. Cells were fixed in 4% Paraformaldehyde for 20 min followed by Hoechst (1:2000) staining for DNA and mounting in Prolong Gold Antifade Mountant. Images were taken and quantified on Biotek Lionheart Automatic Microscope. Representative data are shown for experiments repeated three of more times with consistent results.

## Flow cytometry and cell cycle analysis

Click-iT RNA Alexa Fluor Imaging Kit was adapted to label newly synthesized RNAs in the cells for flow cytometry. Abl pre-B cells grown in a 24-well plate were irradiated with 10 Gray and allowed to recover at different times. 2 mM EU was added to the medium and incubated for 30 min for EU incorporation. Cells were then washed with PBS, fixed in 4% paraformaldehyde PBS for 15 min at room temperature, and permeabilized in cold 0.5% TritonX-100 PBS for 5 min. Click-iT reaction cocktail was prepared according to the manufacturer's protocol and immediately added to the cells to perform click reaction in the dark for 30 min at room temperature. Cells were then washed with Click-iT reaction rinse buffer

(Component F) and 3% BSA-PBS, respectively. For cell cycle analysis, BrdU (10 ug/mL) was added to the cells and incubated for 30 min to label new DNAs. Cells were washed with PBS, fixed in 4% para-formaldehyde PBS for 15 min at room temperature, and permeabilized in cold 0.5% Triton X-100 PBS for 5 min. Cells were then digested with DNase (BD Biosciences, 51-2358KC) for 1 hr at 37 °C. Subsequently, cells were incubated with Alexa Fluor 488 Mouse anti-BrdU (BD Biosciences, 51–9004981, 1:500) in 3% BSA-FBS for 1 hr at room temperature and washed 2 x with PBS, followed by FxCycle Violet (Thermo Scientific, R37166) or 7-AAD (BD Pharmingen, 559925) staining for 10 min. Cells were resuspended in PBS and analyzed on BD LSRII Flow Cytometer or BD LSRFortessa Flow Cytometer. Flow cytometry results were further analyzed using FlowJo software. Representative data are shown for experiments repeated three of more times with consistent results.

## CRISPR-Cas9 screen

More than 140 million wild-type Abl pre-B cells carrying inducible Cas9 transgene were transduced with a lentiviral gRNA library containing 90,230 gRNAs targeting over 18,000 mouse genes (Addgene, 67988) by spin-infection as described above. 3 days post-infection, cells transduced with gRNAs were sorted on a BD FACSAria II Cell Sorter based on BFP expression. BFP-positive cells were treated with 3 µg/ml doxycycline for 7 days to induce gRNA expression and gene inactivation. Cells were irradiated with 10 Gray, allowed to recover for 30 min, processed as described above for EU labeling of newly synthesized RNAs, and analyzed on BD FACSAria II Cell Sorter. Cells with high (top 10%) EU staining and unsorted cells were collected, and the genomic DNA of the cells were isolated for library preparation using nested-PCR. The library was sequenced on an Illumina HiSeq 2500 platform. Raw fastq files were demultiplexed by the Genomics and Epigenomics Core Facility of the Weill Cornell Medicine Core Laboratories Center. The gRNA sequence region was then retrieved from the sequencing data using Seqkit *Shen et al., 2016* and mapped to the gRNA sequence library (*Koike-Yusa et al., 2014*; *Tzelepis et al., 2016*). The number of reads of each library sequence was counted and then normalized as follows *Shalem et al., 2014*. Normalized reads per gRNA = reads pers gRNA total reads for all sgRNAs in sample $\times 10^6$ + 1. Hereby, the generated normalized reads from each guide RNA were used and compared between the EU high cell and unsorted cell. P values were measured by the Poisson test to compare guide RNAs between EU high cells and unsorted cells. FDR was used for adjusting the p-value. CRISPR score = log2 (final sgRNA abundance/initial sgRNA abundance). The EU high genes were defined as these genes that have at least one guide RNA with p adjust value ≤ 0.01 & FC ≥ 1.5. Gene ontology (GO) analysis was performed by the R package cluster Profiler v3.18.1.

## Isolation and deep sequencing of EU-labeled nascent transcripts

Click-iT Nascent RNA Capture Kit (Thermo Fisher, C10365) was used to label and capture the nascent transcripts. In brief, the same number of Abl pre-B cells was plated in two T-25 Polystyrene flasks, one for irradiation (10 Gray) and the other for no IR control. After irradiation, both flasks of cells were recovered for 30 min and incubated in a medium with 2 mM EU for 30 min to allow for the incorporation of EU into the nascent transcripts. Total RNAs were harvested using TRIzol reagent (Thermo Fisher, 15596018) following the manufacturer's protocol. The click-iT reaction was performed as per the manufacturer's protocol in 50 µL total volume for 30 min in the dark. Subsequently, 1 µL ultrapure glycogen, 50 µL 7.5 M ammonium acetate, and 700 µL of chilled 100% EtOH were added to the reaction. The mixture was incubated at −80 °C for 16 hr. RNA pellet was spun down at 13000×g for 20 min at 4 °C, washed 2 x with chilled 75% EtOH, and resuspended in nuclease-free water. EU-RNAs were pulled down with Dynabeads MyOne Streptavidin T1 magnetic beads and extracted with TRIzol reagents. The same amount of ERCC spike-ins (Thermo Fisher, 4456740) were added to the purified EU-RNAs, followed by cDNA library generation using NEBNext Ultr II Directional RNA Library Prep Kit for Illumina (NEB, E7760) according to the manufacturer's protocol and deep sequencing on Illumina HiSeq 2500 platform.

## Real-time quantitative RT-PCR of EU-labeled nascent transcripts

EU-labeled RNA was prepared from the same number of cells as described above. All the isolated EU-RNAs were used for cDNA synthesis using Superscript III (Thermo Fisher, 18080044) reverse transcription with random hexamer as primers following the manufacturer's protocol. The same proportion of cDNA products in each sample was used as a template for the quantitative RT-PCR reaction with

Light Cycler 480 SYBR Green I Master Mix (Roche, 04707516001). The Ct value was used to represent the absolute amount of EU-RNAs in each sample, in which a smaller Ct value indicates higher nascent transcript levels of an individual gene, given the same initial number of cells and the same proportion of EU-RNAs were used for the analyses. Primer sequences for the analyses are as follows:

ms28S-fwd, 5'-TGGGTTTTAAGCAGGAGGTG-; ms28S-rev, 5'-GTGAATTCTGCTTCACAATG -3'(*Watada et al., 2020*); ms18S-fwd, 5'-CTTAGAGGGACAAGTGGCG-3'; ms18S-rev, 5'-ACGCTGAG CCAGTCAGTGTA-3' (*Stephens et al., 2011*); msHist1h2ab-fwd, 5'-GCCTGCAGTTCCCCGTA-3'; msHisth2ab-rev, 5'- ATCTCGGCCGTCAGGTACTC-3'; msHist1h2ac-fwd, 5'-GGCTGCTCCGCAAGGG T-3'; msHist1h2ac-rev, 5'-CTTGTTGAGCTCCTCGTCGTT-3'; msH2afz-fwd, 5'- ACTCCGGAAAGG CCAAGACA-3'; msH2afz-rev, 5'-GTTGTCCTAGATTTCAGGTG-3' (*Nishida et al., 2005*); msCdk-n1a-fwd, 5'-GTGGCCTTGTCGCTGTCT-3'; msCdkn1a-rev, 5'-TTTTCTCTTGCAGAAGACCAATC-3' (*Béguelin et al., 2017*).

## Analysis of nascent transcripts

The mouse genome version GRCm38.p6 release M23 and the associated GENCODE version of the mouse reference gene set were downloaded from the GENCODE website (https://www.gencode-genes.org/mouse/release_M23.html). We trimmed adapter sequences and low-quality sequences in RNA-seq data using the Trim Galore v0.6.6 (*Martin, 2011*) with default parameters. To avoid rRNA homologous sequences (i.e., in the intron regions of Zc3h7a or Cdk8) prior to subsequent genomic and other RNA analysis, we first mapped the reads to mm10 rDNA sequences by TopHat v2.1.1 (*Kim et al., 2013*). The unmapped reads were then further mapped to the mouse genome version mm10 and ERCC spike-in version ERCC92 using TopHat v2.1.1 (*Kim et al., 2013*). Successfully mapping reads were sorted by SAMtools v1. 5. Afterward, read counts in several types of genomic feature, i.e., protein-coding genes, rDNA, and ERCCs (ERCC92.gtf), were quantified by Htseq-count v0.11.2 (*Anders et al., 2015*) using the union gene region option. The read number per gene was normalized based on the total ERCC read numbers in each sample.

To visualize read coverage across the genome, DeepTools v3.5.0 (*Ramírez et al., 2014*) was used to convert BAM files into bigwig files using scale factors calculated by the total ERCC read number in each sample. Next, DeepTools was used to plot the average read depth per sample across inter-ested groups of genomic regions (i.e. repressed protein-coding genes from 3 kb upstream to 3 kb downstream of gene bodies). Screenshots of read density at individual regions were generated by IGV 2.8.13 (*Thorvaldsdóttir et al., 2013*).

We then used one tail Poisson test to evaluate the difference in gene expression level based on the read counts normalized by total ERCC read counts. We defined differentially expressed RNAs as those with a fold change greater than 1.5 and an FDR value smaller than 0.05. To detect highly expressed genes, we ranked genes by RPKM in the control cells, whereas RPKM was calculated using ERCC-normalized read counts further normalized by gene length. GO analysis was performed by the R package clusterProfiler v3.18.1 (*Wu et al., 2021*). Heatmap were generated by pheatmap.

## Acknowledgements

The authors thank Yinan Wang for performing the bioinformatics for the high throughput screen. We thank the Weill Cornell Flow Cytometry Core for flow cytometry. We thank the Weill Cornell Epig-enomics Core for performing the sequencing for the high throughput screen and the Transcriptional Regulation & Expression Facility at Cornell University, Ithaca for providing advice and performing the sequencing of the nascent transcripts. The stable doxycycline-inducible eGFP-TOPBP1 U2OS cell lines were a kind gift from Dr. Helmut Pospiech (Fritz Lipmann Institute, Germany). We are grateful to Pengbo Zhou for advice on Cullin 4 A and B We also thank Barry Sleckman, Bo-Ruei Chen, and Faith Fowler for advice throughout the project. JKT is supported by NIH R35 GM139816 and RO1 CA95641. KC is supported by NIH R01GM138407, R01GM125632, R01HL148338, and R01HL133254.

## Additional information

### Funding

| Funder | Grant reference number | Author |
|---|---|---|
| National Institutes of Health | GM139816 | Jessica K Tyler |
| National Institutes of Health | CA95641 | Jessica K Tyler |
| National Institutes of Health | GM138407 | Kaifu Chen |
| National Institutes of Health | GM125632 | Kaifu Chen |
| National Institutes of Health | HL148338 | Kaifu Chen |
| National Institutes of Health | HL133254 | Kaifu Chen |

The funders had no role in study design, data collection and interpretation, or the decision to submit the work for publication.

### Author contributions

Zulong Chen, Data curation, Formal analysis; Xin Wang, Xinlei Gao, Nina Arslanovic, Formal analysis; Kaifu Chen, Supervision, Writing – review and editing; Jessica K Tyler, Conceptualization, Project administration, Writing – review and editing

### Author ORCIDs

Zulong Chen (iD) http://orcid.org/0000-0002-5423-1264
Xin Wang (iD) https://orcid.org/0000-0003-1456-335X
Kaifu Chen (iD) http://orcid.org/0000-0003-1009-4357
Jessica K Tyler (iD) https://orcid.org/0000-0001-9765-1659

Reviewer #1 (Public review): https://doi.org/10.7554/eLife.94001.3.sa1
Reviewer #2 (Public review): https://doi.org/10.7554/eLife.94001.3.sa2
Author response https://doi.org/10.7554/eLife.94001.3.sa3

## Additional files

### Supplementary files

• Supplementary file 1. Nascent RNA profiles of each gene using ethynyl uridine (EU) RNA-seq.

• Supplementary file 2. Significantly enriched Gene ontology (GO) terms for up-regulated gene after irradiation.

• Supplementary file 3. Significantly enriched Gene ontology (GO) terms for down-regulated genes after irradiation.

• Supplementary file 4. Whole genome CRISPR-Cas9 screen detects the abundance of all gRNAs and target genes for ethynyl uridine (EU) high cells and unsorted cells.

• MDAR checklist

### Data availability

All raw sequencing data from the nascent EU RNA-seq and CRISPR screen experiments have been deposited in the NCBI project database under accession PRJNA895065. The genome-wide gRNA library CRISPR-Case9 screen datasets comprise Abl pre-B cells of both unsorted (SRX18076832) and sorted the 10% of the cells with the most nascent RNA (high EU) 30 minutes after IR (SRX18076831). The raw FASTQ files for nascent EU RNA-seq include pre-B cells without irradiation (SRX18050529 and SRX18050531) and with 30 minutes after irradiation (SRX18050530 and SRX18050532). Additionally,

both FASTQ files and processed data for nascent EU RNA-seq are accessible at GSE217123. The source code employed in the data analysis and figure generation have been uploaded to GitHub at the following repository: https://github.com/gucascau/NascentDiff (copy archived at *Wang, 2024*).

The following datasets were generated:

| Author(s) | Year | Dataset title | Dataset URL | Database and Identifier |
| --- | --- | --- | --- | --- |
| Chen Z, Wang X, Gao X, Arslanovic N, Chen K, Tyler J | 2024 | Transcriptional inhibition after irradiation occurs preferentially at highly expressed genes in a manner dependent on cell cycle progression | https://www.ncbi.nlm.nih.gov/bioproject/PRJNA895065 | NCBI BioProject, PRJNA895065 |
| Chen Z, Wang X, Gao X, Arslanovic N, Chen K | 2024 | Transcriptional inhibition after irradiation occurs preferentially at highly expressed genes in a manner dependent on cell cycle progression | https://www.ncbi.nlm.nih.gov/geo/query/acc.cgi?acc=GSE217123 | NCBI Gene Expression Omnibus, GSE217123 |

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
