## [Editor Report · eLife assessment]

This **important** work describes a **compelling** analysis of DNA damage-induced changes in nascent RNA transcripts, and a genome-wide screening effort to identify the responsible proteins. A significant discovery is the inability of arrested cells to undergo DNA damage-induced gene silencing, which, is attributed to an inability to mediate ATM-induced transcriptional repression. This work will be of general interest to the DNA damage, repair, and transcription fields, with a potential impact on the cancer field.

---

## [Referee Report · Reviewer #1 (Public review)]

This manuscript by Tyler and colleagues describes a thorough analysis of IR-induced changes in nascent RNA transcripts, and a genome-wide screening effort to identify the responsible proteins. The findings extend previous work describing DNA damage-induced transcriptional repression from DNA breaks in cis to bulk genomic DNA damage. A significant discovery is the inability of arrested cells to undergo DNA damage-induced gene silencing, which, at least at the rDNA locus, is attributed to an inability to mediate ATM-induced transcriptional repression. While the findings add to our knowledge of how DNA damage affects gene expression, there are several limitations to the current study that remain inadequately addressed. In addition, some of the proposed conclusions seem speculative and should be marked as such, omitted or experimentally supported.

Two major concerns were as follows and have been addressed as outlined in the authors' response to this review:

(1) The CIRSPR screen designed to detect regulators of damage-induced transcriptional repression is based on EU incorporation following a 7-day selection of stable knockout cells. As the authors point out, cell cycle arrest reduces rDNA transcription on its own. The screen, which assesses changes in sgRNA distribution in EU high cells, is thus likely to be dominated by factors that affect cell cycle progression. This is exemplified in the analyses of top hits related to neddylation. The screen's limitations in terms of identifying DDR effectors of damage-induced silencing needs to be clearly stated.

(2) The authors confirm previous findings of DNA damage-induced repression of rDNA and histone gene transcription. The authors propose that these highly transcribed genes are more susceptible to silencing than the bulk of protein coding genes and propose a global damage-induced signaling event that is independent of DNA breaks in cis. While this is possible, it is not demonstrated in this manuscript, and the authors should acknowledge alternative explanations. For example, the loci found to be repressed by bulk IR are highly repetitive gene arrays that tend to form nuclear sub-compartments (nucleoli, histone bodies). As such, their likelihood of being in the vicinity of DNA damage is high, at least for a fraction of gene copies. The findings, therefore, remain consistent with cis-induced silencing. Moreover, silencing may spread through the relevant nuclear sub-compartments, consistent with the formation of DNA damage compartments described recently (PMID: 37853125).

Other comments - also addressed in the authors' response:

(1) The statement that silencing is due to transcription initiation rather than elongation is not sufficiently supported by the data. Could equivalent nascent transcript reduction not be the result of the suppression of elongating RNA PolII? To draw the proposed conclusion, the authors would need to demonstrate that RNA PolII initiation is altered, using RNA PollII ChIP and/or analysis of relevant RNA PolII phosphorylation patterns.

(2) The lack of rDNA silencing in arrested cells is interesting, though the underlying mechanism remains unclear. To further corroborate the proposed defect in ATM-mediated signaling, the authors should look directly at ATM and Treacle phosphorylation upstream of TOPBP1.

(3) The "change in relative heights of the EU low (G1) and EU high (S/G2) peaks" in Figs 5D, 5E and 6B is central to the proposed model of transcriptional changes being affected by cell cycle arrest. These differences should be visualized more clearly and quantified across independent experiments. Ideally, cell cycle stage should be dissected as in Fig. 2B. How do the authors envision cell cycle arrest triggers the defect in transcriptional silencing?

---

## [Referee Report · Reviewer #2 (Public review)]

In this manuscript, the authors attempted to study mechanisms of transcription inhibition in cells treated with IR. They observed that unlike transcription inhibition induced by UV damage that depends on histone chaperone HIRA, IR induced transcription inhibition is independent on HIRA. Through a CRISPR/Cas9 screen, they identified protein neddylation is important for transcription inhibition. By sequencing nascent RNA, they observed that down-regulated transcripts upon IR treatment are largely highly transcribed genes including histone genes and rDNA.

This study utilized comprehensive approaches to fill in knowledge gap of IR-induced transcription inhibition.

Comments on current version:

The revised manuscript largely addressed my concerns.

---

## [Author Response]

The following is the authors’ response to the original reviews.

**Reviewer 1:**
Comment (1) The CIRSPR screen designed to detect regulators of damage-induced transcriptional repression is based on EU incorporation following a 7-day selection of stable knockout cells. As the authors point out, cell cycle arrest reduces rDNA transcription on its own. The screen, which assesses changes in sgRNA distribution in EU high cells, is thus likely to be dominated by factors that affect cell cycle progression. This is exemplified in the analyses of top hits related to neddylation. The screen's limitations in terms of identifying DDR effectors of damage-induced silencing need to be clearly stated.

Notably, our screen did identify known DNA damage response effectors of damage-induced silencing, for example ATM was a top hit, as discussed in the paper and shown in Fig. 5B. We consider that our unbiased approach had advantages because in addition to finding known DDR effectors, we uncovered novel requirements, such as the need for cells to be cycling, for transcriptional silencing in response to DNA damage. We didn’t find the canonical key cell cycle regulators in our screen. One possibility might be that cell cycle arrest or cell death upon their knock down may lead to out-competition during the seven-day treatment with doxycycline resulting in depletion from, rather than enrichment in, the targeting gRNAs from cells that maintain transcription 7 days after DNA damage.

Comment (2) The authors confirm previous findings of DNA damage-induced repression of rDNA and histone gene transcription. The authors propose that these highly transcribed genes are more susceptible to silencing than the bulk of protein-coding genes and propose a global damage-induced signaling event that is independent of DNA breaks in cis. While this is possible, it is not demonstrated in this manuscript, and the authors should acknowledge alternative explanations. For example, the loci found to be repressed by bulk IR are highly repetitive gene arrays that tend to form nuclear sub-compartments (nucleoli, histone bodies). As such, their likelihood of being in the vicinity of DNA damage is high, at least for a fraction of gene copies. The findings, therefore, remain consistent with cis-induced silencing. Moreover, silencing may spread through the relevant nuclear sub-compartments, consistent with the formation of DNA damage compartments described recently (PMID: 37853125).

The reason for us “suggest(ing) that the reduced bulk abundance of nascent transcripts after IR may occur in trans as a programmed event” was based on the gene length-independent and IR dose-independent nature of the gene silencing shown in Fig. 2D and Fig. 4C, not that rDNA and histone gene expression went down the most after IR. Indeed, we stated that “Those genes that were normally most highly transcribed were repressed after IR, while genes that were normally expressed at intermediate or low levels tended to be induced after IR (Fig. 4A). The mechanistic reason for this is unclear.” We thank the reviewer for the suggestion that this may be due to these genes existing in nuclear sub-compartments. We have now incorporated this possibility into the discussion.

Other comments:(1) The statement that silencing is due to transcription initiation rather than elongation is not sufficiently supported by the data. Could equivalent nascent transcript reduction not be the result of the suppression of elongating RNA PolII? To draw the proposed conclusion, the authors would need to demonstrate that RNA PolII initiation is altered, using RNA PollII ChIP and/or analysis of relevant RNA PolII phosphorylation patterns.

Figure 4F shows the distribution of nascent transcript reads throughout the open reading frame of the repressed genes. It shows that the transcript abundance throughout the ORF, including at the 5’ end, is reduced. This pattern is consistent with a defect in initiation. We have now clarified the description of these results to state that: “Our data is consistent with the possibility that the major mechanism for the repression of the ~1,000 protein coding genes after IR is at the transcriptional initiation stage. However, our data do not rule out that elongation may be additionally repressed after IR, as this would not be observed in our analyses due to concomitant repression of transcriptional initiation.”

(2) The lack of rDNA silencing in arrested cells is interesting, though the underlying mechanism remains unclear. To further corroborate the proposed defect in ATM-mediated signaling, the authors should look directly at ATM and Treacle phosphorylation upstream of TOPBP1.

We would love to have shown that ATM dependent phosphorylation does not occur upon IR. We had attempted this multiple times but unfortunately the available phospho Treacle antibodies were not suitable for rigorous analyses in our hands.

(3) The "change in relative heights of the EU low (G1) and EU high (S/G2) peaks" in Figures 5D, 5E, and 6B is central to the proposed model of transcriptional changes being affected by cell cycle arrest. These differences should be visualized more clearly and quantified across independent experiments. Ideally, the cell cycle stage should be dissected as in Figure 2B. How do the authors envision cell cycle arrest triggers the defect in transcriptional silencing?

In the previous version, the last paragraph described one possibility for how rDNA may fail to be repressed in arrested cells after IR, based on the results shown in Fig. 7F and G. We have now added a paragraph in the discussion section beginning “Why would cell cycle arrest in G1 or G2 phases of the cell cycle prevent transcriptional repression of rDNA and histone genes after IR?”

**Reviewer #2:**
(1) Define ERCC normalization.

We apologize for this omission. We now have explained ERCC normalization and have added a citation to a commentary that we wrote on spike-in controls 2015 for further explanation.

(2) On page 8, the authors speculate that genes involved in immune response after IR was activated due to cytoplasmic DNA in pre-B cells. Where are these cytoplasmic DNAs from? Is there any literature indicating that 30 30-minute IR treatment can induce cytoplasmic DNA?

We have removed this speculation, as there is no evidence currently to support it.

(3) Related to the points above, are ERVs or repetitive DNA elements up-regulated upon IR treatment, which in turn results in increased expression of genes involved in immune response?

The induction of cytokines as a rapid response to irradiation is a major part of the immediate early gene program induced in response to ROS (and now is explained in the manuscript).

(4) Please explain in the result section how overlap levels of transcription determined by EU are reduced after IR, and yet the number of genes with increased expression upon IR treatment is much more than that of genes with reduced expression.

We have explained that while less genes have reduced expression after IR than the number of genes that increase expression after IR, those genes that have reduced expression are extremely highly expressed to start off with. As a result, the bulk amount of transcripts is reduced after IR.

(5) Do cells treated with MLN4924 block the down-regulation of histone genes and ribosomal genes?

We have not addressed this directly. However, given that the reduction of gene expression that occurs after IR is largely due to repression of histone and rDNA genes, it is safe to speculate that these are the genes that are no longer repressed during cell cycle arrest.

(6) Is IR-induced down-regulation of histone genes due to cell cycle changes?

We do not know for sure if this is the case. It is relevant to note that even without IR, histone expression per se is regulated by cell cycle changes, being lower outside of S phase – and the majority of non-arrested cells in our study are in S phase (Fig. 2B). As such, arrest of cells per se outside of S phase would be sufficient to reduce histone expression level.

We would like to thank the reviewers again for their insightful suggestions and comments.